# Competitive hierarchies in bryozoan assemblages mitigate network instability by keeping short and long feedback loops weak

Franziska Koch [1,2,5✉], Anje-Margriet Neutel[3,5], David K. A. Barnes [3], Katja Tielbörger[1], Christiane Zarfl [1] & Korinna T. Allhoff [1,2,4✉]

Competitive hierarchies in diverse ecological communities have long been thought to lead to instability and prevent coexistence. However, system stability has never been tested, and the relation between hierarchy and instability has never been explained in complex competition networks parameterised with data from direct observation. Here we test model stability of 30 multispecies bryozoan assemblages, using estimates of energy loss from observed interference competition to parameterise both the inter- and intraspecific interactions in the competition networks. We find that all competition networks are unstable. However, instability is mitigated considerably by asymmetries in the energy loss rates brought about by hierarchies of strong and weak competitors. This asymmetric organisation results in asymmetries in the interaction strengths, which reduces instability by keeping the weight of short (positive) and longer (positive and negative) feedback loops low. Our results support the idea that interference competition leads to instability and exclusion but demonstrate that this is not because of, but despite, competitive hierarchy.

[1] University of Tübingen, Tübingen, Germany. [2] University of Hohenheim, Stuttgart, Germany. [3] British Antarctic Survey, Cambridge, UK. [4] KomBioTa – Center for Biodiversity and Integrative Taxonomy, University of Hohenheim & State Museum of Natural History, Stuttgart, Germany. [5] These authors contributed equally: Franziska Koch, Anje-Margriet Neutel. ✉email: franziska.koch@uni-hohenheim.de; korinna.allhoff@uni-hohenheim.de

The coexistence of many competing species is a fundamental challenge for ecological theory. Classic coexistence theory of interference competition explains that two species will be able to return to their steady state after a small perturbation if intraspecific competition outweighs interspecific competition: $a_{ii}a_{jj} > a_{ij}a_{ji}$[1–3], where $a_{ij}$ is the effect of species $j$ on $i$, and all $a_{ij}$ are negative. This can be understood in terms of a balance in dynamic forces: The interaction between two species generates a self-reinforcing feedback loop whose strength is quantified as the product $a_{ij}a_{ji}$. This positive feedback must be counteracted by at least an equally strong self-damping feedback from intraspecific competition ($a_{ii}a_{jj}$) in order to obtain stability. In complex competition networks, there are many more feedback loops, positive and negative.

It has been suggested that dense "intransitive" networks of interactions between many competing species can prevent competitive exclusion and enable coexistence[4]. The claim is that a strict hierarchy has a destabilising effect[5–7] but see ref. [8], while "competitive reversals"[9] enhance the stability of competitive communities[10–12] through negative feedback loops[13]. This is analogous to a rock-paper-scissors game, where there is no superior competitor[14].

However, many multispecies natural systems show a clear ranking from strongest to weakest competitor, in contrast to what intransitivity theory would suggest[8,15–18]. Furthermore, theoretical studies on the relation between intransitive competition and stability have largely been based on very simple "tournament-style" competition models[7,9,19]. Also, the strengths of intraspecific competition have been largely ignored[19]. As for empirical studies, these are usually done with pairwise experiments, where pairs of species are grown in isolation to determine competitive dominance (see, for example, refs. [8,17,20]). The relation between competitive hierarchy or intransitivity and instability has thus not been explained in complex dynamic-system models parameterised from whole-community observations of interference competition.

For a two-species interference-competition system, it is easy to demonstrate how the dominance of one species over the other translates into an asymmetry in interaction strengths, which affects the stability of the system. The more asymmetric the division in strength between the two competitors, the weaker the positive feedback will be, and thus the less intraspecific interaction will be needed for system stability (Fig. 1). Also, in larger systems self-damping feedback must, at least, compensate for the total positive 2-link feedback from the interacting pairs[21]. A strong hierarchy and hence asymmetry in pairwise interactions will keep the 2-link feedback loops relatively weak, which would suggest that relatively little self-damping is needed to make asymmetric networks stable.

This raises two questions: Are such natural interference-competition systems stable, that is, does intraspecific competition outweigh interspecific competition? And second, how does the hierarchical organisation of competitive strengths affect system stability?

Here, we model the stability of 30 assemblages of shallow coastal bryozoans from the Arctic and Antarctic regions. These regions are known for their destructive environmental disturbance events, such as ice scour, which have been shown to cause hierarchical patterns of competition in benthic communities[15]. Bryozoans are sessile, aquatic, colonial, suspension-feeding animals (Fig. 2A). If bryozoan colonies grow into each other, it leads to overgrowth of one colony by the other (win/loss outcome) or mutual overgrowth or cessation of growth at the boundary (draw). Competition leads to energy loss for each competitor, whatever the outcome.

We observed the spatial contests between the colonies in the assemblages and estimated the energy loss rates resulting from

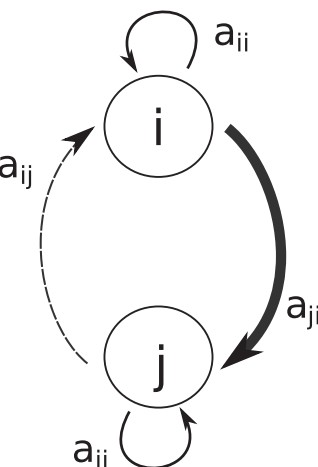

**Fig. 1 Asymmetry, feedback and stability in a 2-species system.** Competition creates a positive feedback loop ($a_{ij}a_{ji}$), where $a_{ij}$ and $a_{ji}$ are both negative. For a 2-species competition system to be stable, this positive feedback must be counterbalanced by the same amount of negative feedback caused by the two self-regulation effects $a_{ii}a_{jj}$. Asymmetry between the two competitors reduces the product of $a_{ij}a_{ji}$ and with that the amount of self-regulation that is needed to obtain stability.

each contest, constructing "energy loss webs" in analogy to energy flow webs in food-web theory[22]. We used these webs to calculate the species' per-capita interaction strengths sensu May[23] (Fig. 2B). The interaction strengths are the elements of the Jacobian matrix, the partial derivatives of the species' growth equations at steady state. We made the steady state assumption to be able to test the stability of the networks since, in order for a system to be stable, the growth and loss rates of each species in a community must be in balance. Note that the concept of stability in our study is that of local system stability and does not refer to the stability of the environment nor to the ability of the system to deal with any specific substantial outside disturbance. Local stability refers to the internal balance in the dynamic forces in the network and is the ability of the system to return to the steady state after an infinitesimal perturbation of this state[23].

With our dynamic-system analysis of the 30 observed bryozoan assemblages, we aimed not only to test whether the competition networks are stable but also to provide a fundamental understanding of the relation between hierarchy and stability. We did this by quantifying the system's feedback structures which allowed us to compare and explain the stability results. Finally, we synthesised our findings on interference-competition ($-/-$ interactions) with stability analyses of trophic networks ($+/-$ interactions)[23–26] and discuss a general organising feedback principle for ecological networks.

## Results

**Hierarchy and asymmetry in observed competition networks.** The species' competition networks, ranging in size from 5 to 11 species, were highly connected, with most connectance values around 0.9 ("Methods", Supplementary Table 1). In all networks, for almost all pairs of interacting species, we found that competition was characterised by a clear dominance of the stronger competitor over the weaker competitor in terms of the relative number of contests won by colonies of the species. This dominance could be described by Tanaka and Nandakumar's win index[27], which ranged from 0.63 to 0.96 (Supplementary Table 1). Furthermore, for all networks, species could be ranked into perfect or almost perfect competitive hierarchies, with each

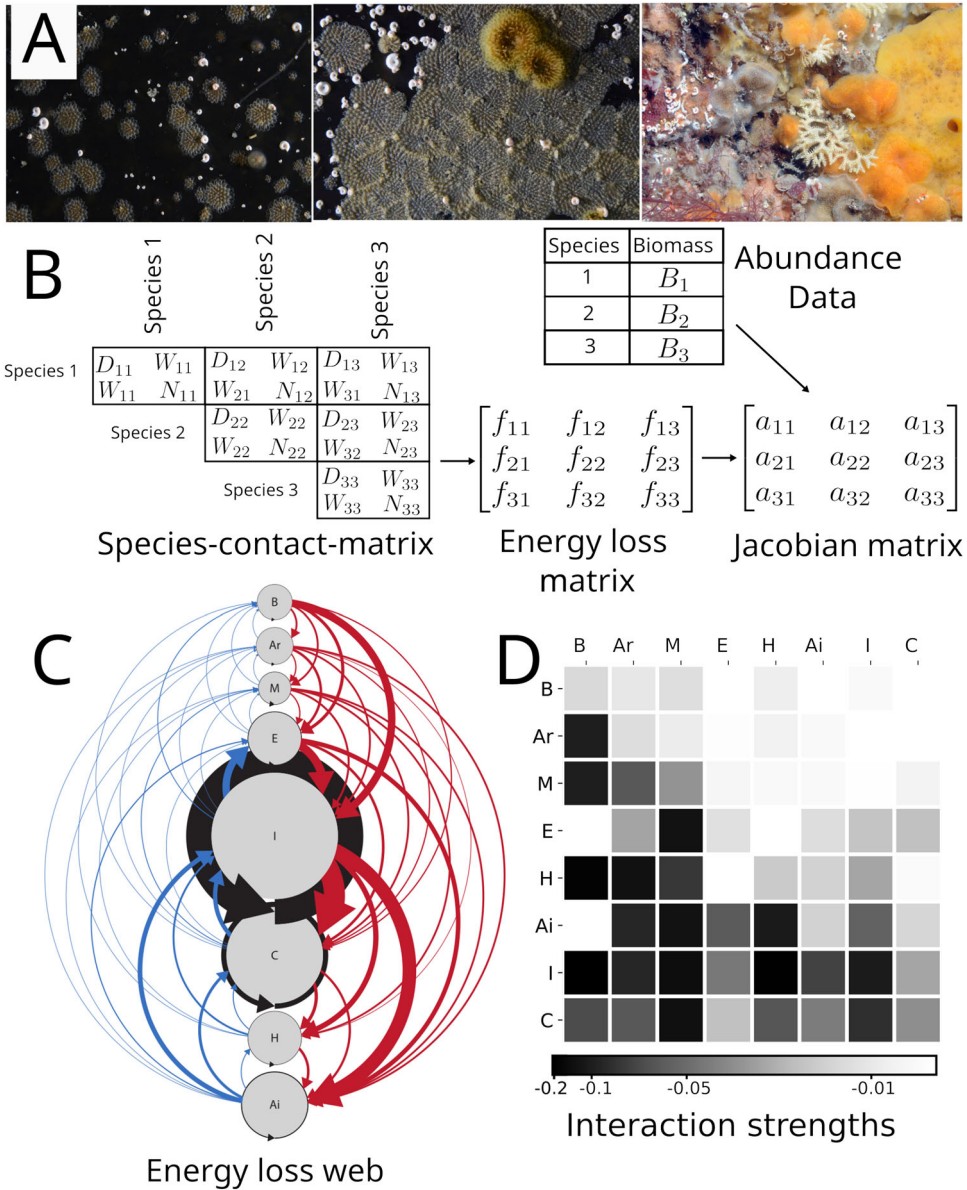

**Fig. 2 Translating observations of competitive interactions into interaction strengths. A** Examples of bryozoan assemblages (photos by: British Antarctic Survey, Cambridge, UK). **B** Schematic overview of our data processing routine. Species-contact matrices summarise observed competitive outcomes between the colonies of pairs of species (number of draws (D), wins (W) and total number of confrontations (N) between colonies). These are translated into species' energy loss rates $f_{ij}$ (biomass per time) by weighing the cost of winning, losing and drawing. Per-capita interaction strengths, the elements of the Jacobian matrix, are calculated by combining energy loss rates with abundance data (see "Methods"). **C** Example energy loss web from Signy Island, Antarctica, containing eight species (B= *Beania erecta*, Ar = *Arachnopusia inchoate*, M = *Micropora brevissima*, E = *Escharoides tridens*, I = *Inversiula nutrix*, C = *Celleporella bougainvillea*, H = *Hippadanella inerma* and Ai = *Aimulosia antarctica*). Node size indicates the observed abundance of each species. The thickness of an arrow pointing from species *j* to species *i* indicates the amount of energy that species *i* loses per time due to interference competition with species *j*. The species are organised hierarchically, from the strongest to weakest competitor in terms of energy loss, with red arrows representing "top-down" losses and blue arrows representing "bottom-up" losses. The hierarchy causes pairwise asymmetry in each coupled pair of interaction strengths, as well as community asymmetry with strong "top-down" and much weaker "bottom-up" loss rates. **D** The observed asymmetric patterns are even more pronounced in the interaction strengths (Supplementary Table 1). The community pattern appears here as a clear above-below diagonal asymmetry.

competitor outcompeting all or most species below it (Supplementary Fig. 1). This is in line with Gallien's intransitivity index[28], which resulted in values ranging from −0.3 to about 0.1 when applied to our data, demonstrating that our networks were indeed very transitive (Supplementary Table 1).

We then calculated energy loss rates $f_{ij}$ from observations of interference competition ("Methods"). When we did this, the competitive hierarchy translated into two asymmetric patterns in energy loss. First, for each pair of competitors, one clearly

dominated over the other so that each interaction consisted of a relatively low loss rate coupled with a relatively high loss rate. Second, the relatively strict ranking from strongest to weakest competitor translated into a second, additional asymmetry at the community level with high "top-down" loss rates (losses of lower ranked species caused by competition with a higher ranked species) versus low "bottom-up" rates (Fig. 2C). We quantified both pairwise and community asymmetry (see "Methods") and applied both asymmetry measures to the energy loss rates $f_{ij}$ and

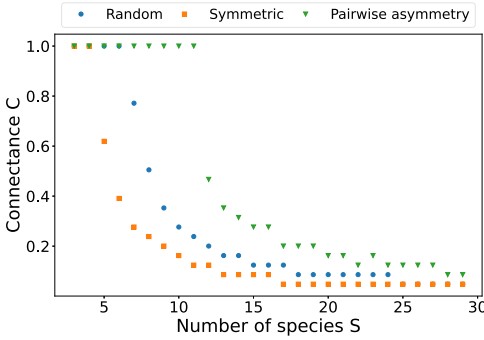

**Fig. 3 Asymmetry and stability in random competition matrices.** In multispecies competition systems, pairwise asymmetry enhances the probability of stability, while symmetry reduces it. For each combination of number of species S and connectance C, we calculated an ensemble of 100 random matrices. Results show the threshold at which less than 5% of the matrices were stable. The data points thus represent systems on the threshold between stability and instability. For a given parametrisation, the stable region is below the curve, and the unstable region is above the curve. All interspecific interaction strengths were drawn from the same normal distribution. Pairwise asymmetry was obtained by coupling strong and weak links to each other, symmetry by coupling strong links to other strong links (see "Methods").

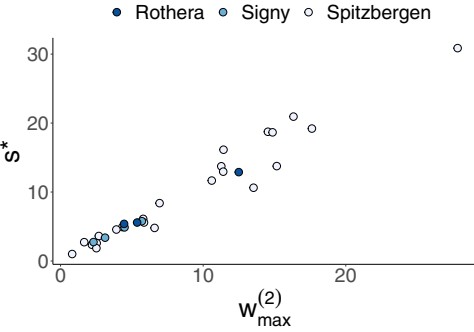

**Fig. 4 Relation between maximum 2-link loop weight $w^{(2)}_{max}$ and critical level of self-regulation s\* in empirical systems.** The critical level of self-regulation s\* represents the factor by which observed intraspecific interaction strength has to be multiplied to obtain stability; s\* > 1 means that a system is unstable, and s\* = 1 represents the threshold between stability and instability. The maximum weight of 2-link loops is denoted as $w^{(2)}_{max}$, where $w^2 = \sqrt{\frac{a_{ij}a_{ji}}{|a_{ii}a_{jj}|}}$. Regression analysis: $n = 30$ bryozoan assemblages, $R^2 = 0.95$, $y = -0.11 + 1.13x$, $p < 0.001$.

to the interaction strengths $a_{ij}$. Both asymmetries were even stronger in the interaction strengths, compared to the energy loss matrices, due to the incorporation of population densities (Fig. 2D, Supplementary Table 1).

**Asymmetry and stability in theoretical competition systems.** We tested the effect of asymmetry on stability in theoretical random networks by arranging randomly drawn interaction strengths to form symmetric or asymmetric pairwise interactions (see "Methods"). While all random networks became more unstable as the number of species S and connectance C were increased, we found that for a given combination of S and C, pairwise asymmetry made the networks more likely to be stable, while pairwise symmetry made them less likely to be stable (Fig. 3).

**Instability of observed competition networks.** We calculated the stability of the observed systems and found that all empirical bryozoan competition networks were unstable; that is, all empirical matrices had dominant eigenvalues with a positive real part (Supplementary Table 1). This meant that intraspecific competition of the species did not provide enough self-damping feedback to counteract the destabilising feedback generated by interspecific competition. We performed a sensitivity analysis to test how extreme scenarios for possible relationships between parameter values affected the stability of the systems. In particular, we varied the energy loss that was allocated to each type of competitive outcome (win/loss/draw). We found that all empirical systems remained unstable in all scenarios and that their relative stability remained unchanged (Supplementary Note 1, Supplementary Figs. 2 and 3).

**Relation between strongest 2-link feedback and comparative instability of observed competition networks.** We then compared the relative instability of the 30 observed networks. In order to enable a meaningful comparison between the different empirical matrices, all with different diagonal values, we determined the critical level (relative to the observed values) of self-

regulation s\* that was needed for stability (following ref. [26], see "Methods" for details). To analyse the stability results, we calculated the "weight" of the feedback loops, following Neutel and co-authors (refs. [24,26], see "Methods").

We found a very strong correlation between the heaviest 2-link positive feedback loop in a competition network, $w^{(2)}_{max}$, and the critical level of self-regulation s\* of the system (see "Methods", Fig. 4). This is interesting, because in 2-species competition systems, $w^{(2)}$, which is called niche overlap by Chesson[29], fully determines system stability. Thus we show that even in more complex systems, stability is governed by the 2-link loops. It is noteworthy that there was no clear correlation between species richness or network complexity and s\* of the observed systems (Supplementary Fig. 4). We also did not find a relation between any of the asymmetry metrics (the transitivity index of ref. [27], pairwise asymmetry or community asymmetry) and s\* of the observed systems (Supplementary Fig. 4).

**Effect of randomised interaction strengths on critical self-regulation.** When we destroyed the asymmetric organisation by randomising the observed patterns of interaction strengths, the systems became even less stable ("Methods", Supplementary Fig. 5). We then normalised the matrices, following Neutel and Thorne[25], in order to enable transparent manipulation experiments with the off-diagonal patterns of interaction strengths, without affecting at the same time the diagonal structure. This preserved the feedback structure and stability properties of the systems ("Methods" and Supplementary Tables 2 and 3, ref. [30]).

In our first manipulation of the normalised community matrices, we randomised all non-zero off-diagonal elements ("Methods") and found again that the systems became less stable. This disruption of patterning tended to weaken the asymmetry within pairs of interacting species, thus causing an increase in the strength of the positive 2-link feedback loops (Supplementary Table 4). We found that pairwise asymmetry was correlated negatively and the strongest 2-link loop $w^{(2)}_{max}$ was correlated positively with critical self-regulation s\* (Fig. 5a, b).

**Relation between hierarchy, feedback structure and stability.** The question remained whether at all, or to what extent, the competitive hierarchies in the networks affected system stability. We examined how the hierarchical structure affected the stability

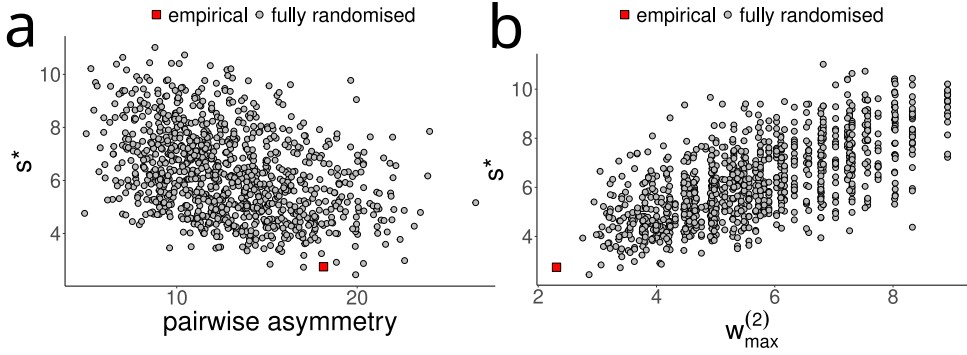

**Fig. 5 Effect of full randomisation on pairwise asymmetry and stability.** Full randomisation destroys the empirical organisation of interaction strengths by reshuffling all non-zero off-diagonal elements within the normalised community matrix. Network topology, complexity and mean interaction strengths are preserved. The randomisation destroys the asymmetry of pairwise interactions (**a**) and increases the maximum weight of 2-link loops $w^{(2)}_{max.}$ (**b**). Both panels show the relationship with stability, quantified as the critical level of self-regulation $s^*$. Data are shown for the web from Signy 1 (Fig. 2C, D) as a representative example. Regression analysis: $n$ for (**a**) and (**b**) = 1001 (1 empirical dataset, 1000 randomised versions) **a** $R^2 = 0.15$, $p < 0.001$, $y = 8.41 - 0.168x$; **b** $R^2 = 0.45$, $p < 0.001$, $y = 1.95 + 0.77x$.

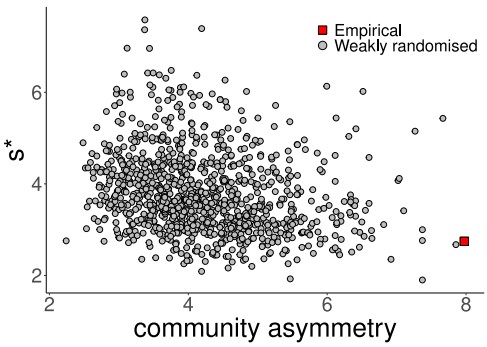

**Fig. 6 Effect of weak randomisation on community asymmetry and stability.** In the weak randomisation, all pairwise interactions are kept intact, but their location is reshuffled within the community matrix. Thus, pairwise asymmetry and all 2-link loops of the empirical matrix are preserved. However, as the location of pairs within the matrix is randomised, community asymmetry is destroyed. Network topology, complexity and mean interaction strength are preserved. Stability is quantified as the critical amount of self-regulation $s^*$. Data are shown for the web from Signy 1 (Fig. 2C, D) as a representative example. Regression analysis: $N = 1001$ (1 empirical dataset, 1000 randomised versions), $R^2 = 0.05$, $p < 0.001$, $y = 4.7 - 0.22x$.

properties of the observed systems by performing a weaker disruption of the patterns in which pairs of non-zero elements were randomly permuted (see "Methods"). This allowed us to separate the effect of pairwise asymmetry from the effect of community asymmetry since all pairwise relationships and their respective asymmetries were preserved in this randomisation.

We found that this weaker disruption also increased $s^*$, be it not as much as with the full randomisation. Higher $s^*$ values in the weakly randomised networks were correlated with lower values of community asymmetry, suggesting a stabilising effect of the hierarchical structure on stability (Fig. 6). However, there was more to the destabilising effect of weak randomisation on $s^*$ than the loss of community asymmetry. This became clear when we performed an even weaker ("minimal") randomisation, where we preserved the pairs of interaction strengths as well as the location above or below the diagonal. This did not affect the community asymmetry but nevertheless disrupted the patterning and also led to an increase in $s^*$.

The effect of weak and minimal randomisation on $s^*$ could be explained by looking at the full spectrum of loop weights (Fig. 7a).

Empirical networks were characterised by a relatively low overall loop weight. Both the weak and the minimal randomisation did not affect 2-link loop weights but increased the weight of all longer—positive and negative—loops, giving an increase in $s^*$. With full randomisation, the 2-link loops also became heavier, which increased $s^*$ even further (Fig. 7b).

To further test this causal relationship between hierarchy and stability, we artificially restored hierarchy in the randomised systems ("Methods"). First, applying an artificial pairwise asymmetry reduced the weight of 2-link loops, leading to a decrease of $s^*$. Adding community asymmetry on top of pairwise asymmetry reduced the weight of longer loops, causing $s^*$ to decrease even further. Thus, hierarchy reduced relative instability through both pairwise and community asymmetry by keeping the overall loop weight low (Fig. 7, see Supplementary Table 4 for the effects of manipulations on stability for all webs).

## Discussion

Interference competition in natural communities is not about winning or losing; it is about losing only. It is a dynamic "loss-loss game" with players interacting in densely, highly connected networks. It has long been thought that competitive hierarchies in such systems are destabilising and that the presence of intransitive loops in competitive networks enables many species to coexist, even with fierce competition between the species[4,7,10,14,31], but see ref. [8]. By modelling loss rates from whole-network observations, we show that, on the contrary, hierarchy reduces instability.

Intransitivity has typically been studied separately from classical coexistence or dynamic-systems frameworks[19]. Theoretical research focuses mostly on game-theoretical tournaments, where interspecific competition is assumed to act in the absence of stabilising mechanisms like intraspecific competition. Furthermore, it is assumed that competitive outcomes are strictly deterministic and that two species cannot be equally strong[7,9]. This makes it challenging to apply traditional hierarchy ideas to empirical systems, which contain interactions of various strengths. We therefore introduced two separate indices of pairwise and community asymmetry to quantify hierarchical patterns of energy loss and interaction strengths in bryozoan assemblages. By distinguishing these two types of asymmetry, we were able to explain the relation between hierarchy, feedback structure and stability.

Although the natural hierarchical organisation of link strengths in the bryozoan networks kept the weight of feedback loops

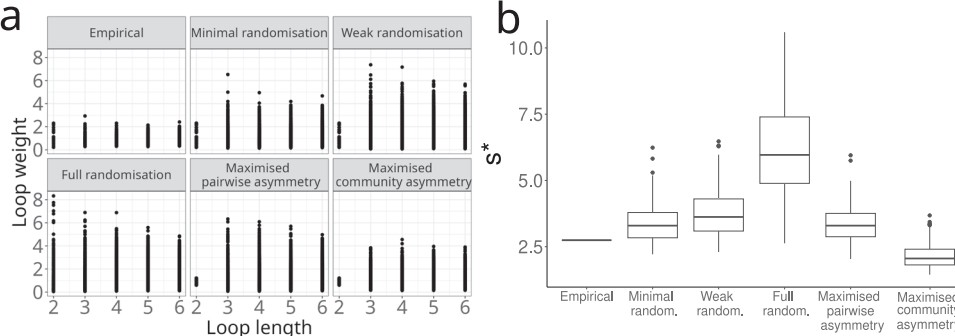

**Fig. 7 Effect of patterns in interaction strengths on relative instability.** The manipulations of the patterns in interaction strengths affect the weight of feedback loops (**a**) and stability (**b**). Loop weights are defined as $w^{(k)} = \left| \frac{a_{i_1 i_2} a_{i_2 i_3} \ldots a_{i_k i_1}}{a_{11} a_{22} \ldots a_{kk}} \right|^{\left(\frac{1}{k}\right)}$, where $k$ is the length of the loop[24]. As all links are negative, loops with an odd number of links are negative (self-dampening), while loops with an even number of links are positive (self-reinforcing). Empirical competition networks are characterised by strong hierarchies in interaction strengths, relatively low $s^*$ and low overall loop weight. Weak randomisation preserves the pairs of interaction strengths and hence keeps the weight of 2-link loops intact but destroys the feedback structure at higher levels of organisation, causing the weight of longer loops to increase and correspondingly causing an increase in $s^*$. Full randomisation is obtained by random permutation of all non-zero off-diagonal interaction strengths. This destroys both pairwise as well as community asymmetry, causing the weight of loops of all lengths to increase, leading to a further increase in $s^*$. Rearranging the randomised matrix to maximise pairwise asymmetry causes 2-link loop weights to decrease strongly, leading to a decrease in $s^*$. Maximising community asymmetry by rearranging the pairs reduces the weight of the longer loops, leading to a further decrease of $s^*$. Boxplots in (**b**) show data from 1000 randomisations with median (centre line), upper and lower quartiles (box limits), 1.5x quartile range (whiskers) and outliers as points. Data are shown for the web from Signy 1 (Fig. 2C, D) as a representative example.

relatively low, measuring intra- as well as interspecific interaction strengths allowed us to show that the bryozoan systems were nevertheless unstable. This is in line with previous findings based on a competitive lawn community[8] and suggests that the systems could be on a trajectory of change. Direct empirical observation[32] indicates that high-latitude bryozoan assemblages are indeed unstable and, if undisturbed, are experiencing competitive exclusion[33,34]. Species richness is maintained by destructive environmental disturbance, e.g., from ice scour, which opens up gaps for recolonisation[35,36]. In our systems, it was the combination of opportunistic colonisers, who dominated in biomass but tended to lose contests, and strong competitors, who won most contests, which caused the hierarchies and provided the asymmetries in interaction strengths[37].

We found that instability was not correlated with system size, in contrast to what is predicted by random matrix theory[23,38]. The same has been shown for empirically parameterised trophic networks[24,25,39], but see ref. [40]. It thus further supports the evidence that the stability of real ecological networks is not determined by their topological structure but by specific non-random patterns in interaction strengths[24–26,39,41–43].

We furthermore found that the relative instability of empirical competition networks was correlated with the maximum strength of the shortest positive feedback loops. This is in line with insights from trophic networks[24,25] and points to a general organising principle for ecological networks. Where in trophic networks these shortest positive loops are 3-link loops[24–26] representing key functional consumer-resource relationships such as intraguild predation[44], in competition networks, these are 2-link loops.

However, to fully understand the effect of hierarchy on the stability of competition networks, we need to go beyond these 2-link loops. We found that community asymmetry, caused by competitive hierarchy, was key in mitigating instability because it kept the negative 3-link and longer loop weight very low. It has indeed long been known that excessive negative feedback from longer loops can destabilise systems via oscillations if it becomes too strong compared to feedback from shorter loops[21].

In conclusion, we found that in all 30 bryozoan assemblages, intraspecific competition did not outweigh interspecific competition; that is, the systems were not stable. We showed that the maximum 2-link positive feedback in the systems governed the relative instability of the systems. Furthermore, we explained how competitive hierarchy translates into pairwise and community asymmetry in energy loss rates and interaction strengths. As they lead to weak overall feedback, these asymmetries reduce system instability. Our findings on the relation between hierarchy and stability may be of relevance not only for ecological research but for network science in general, given that interference competition and hierarchy are common features in many human interaction networks as well[45–47].

## Methods

**Data collection.** Our dataset contains records of overgrowth competition in 30 high-latitude bryozoan assemblages: three from Rothera station on Adelaide Island, West Antarctic Peninsula (67° 34.5' S, 68° 07.0' W, collected between 1997 and 2006)[16], four from Signy Island in the maritime Antarctic (60°43'S, 45°36'W, collected between 1992 and 2006)[48], and 23 from in Spitsbergen, Svalbard in the Arctic (Kongsfjord: (1) 79° 01.8' N, 11° 49.8' E, (2) 78° 59.5' N, 11° 58.9' E, (3) 78° 58.5' N, 11° 29.8' E; Hornsund: (1) 77° 00.8' N, 15° 33.3' E, (2) 76° 56.8' N, 15° 48.4' E, (3) 76° 57.4' N, 15° 55.6' E, collected in 2002)[49]. Analysed communities were limited to bryozoans, and other encrusting species were removed from the dataset. Of the 30 networks, eight have appeared in previous publications (see Supplementary Table 1), while the other 22 have not been published before.

Rocks were collected by hand using SCUBA from shallow subtidal (6–12 m depth) coastal locations. On each rock, all living bryozoan species were identified to species (or morphotype) and individual colonies were counted to give abundance in terms of the number of colonies per species.

**Translating empirical observations into win/loss/draw matrices.** All pairwise contests between colonies were classified as a win, draw or loss. A win for species A and a loss for species B were scored when a colony of species A overgrew 5% of a colony of species B (following ref. [16]). A draw (tie) was scored when two contesting colonies of species A and B showed either cessation of growth along the contact boundary or had equal amounts of mutual overgrowth. The results were compiled in species-contact matrices. The method of data collection is described in more detail in ref. [49].

**Estimating energy loss rates from win/loss/draw observations.** We introduce the concept of "energy loss webs" in analogy to energy flow webs in food-web theory[22]. Energy loss webs describe negative material flow rates $f_{ij}$ (measured as biomass loss per time) for each species $i$ as a result of interaction with species $j$. In our models, we calculated these loss rates as a weighted sum of all the outcomes of observed competitive contests between colonies of species $i$ and $j$. The total energy

loss rate [biomass/time] $f_{ij}$ of species $i$ due to interference competition with species $j$ was calculated as:

$$f_{ij} = p_W W_{ij} + p_L L_{ij} + p_D D_{ij} \qquad (1)$$

where $W_{ij}$ is the number of wins by colonies of species $i$ over colonies of species $j$, $L_{ij}$ the number of colony losses of species $i$ to species $j$ and $D_{ij}$ the number of tied outcomes. The parameters $p_W$, $p_L$, and $p_D$ are constants, representing fixed proportions of energy loss per colony per time for a win, a loss and a draw, respectively. In our study, we assumed that each colony has the same biomass, set as 1, and:

- losing a contest by being overgrown results in an energy loss of $p_L = -0.9$
- winning a contest by overgrowing another colony results in an energy loss of $p_W = -0.1$
- a draw results in an energy loss of $p_D = 0.2$ for both colonies.

In the case of a draw in an intraspecific contest, the number of tied outcomes $D_{ij}$ was doubled as two colonies are affected, but only one contest is recorded in the species-contact matrix. We found that varying the cost values did not have an effect on the qualitative results (see Supplementary Note 1).

**Calculation of interaction strengths.** We modelled the dynamics of $n$-species networks using classic Lotka-Volterra type differential equations: $\frac{dX_i}{dt} = r_i X_i - \sum_{j=1}^{n} c_{ij} X_i X_j$ where $X_i$ is the population density of species $i$, $r_i$ its intrinsic growth rate and functional responses are linear, represented by the coefficient $c_{ij}$, a constant that describes the intensity of competition between species $i$ and $j$.

The dynamics around the equilibrium state $X^*$, where growth and loss terms cancel each other out, are described by the Jacobian matrix A, which contains the partial derivatives of the system, evaluated at $X^*$. The elements $a_{ij}$ of A are called interaction strengths, describing the per-capita effect [dimension 1/t] of a change in the biomass of species $j$ on the biomass of species $i$[23]. Interspecific interaction strengths are $a_{ij} = -c_{ij} X_i^*$. Intraspecific interaction strengths are $a_{ii} = r_i - \sum_{j=1}^{n} c_{ij} X_j^* - c_{ii} X_i^*$. Since at equilibrium $r_i - \sum_{j=1}^{n} c_{ij} X_j^*$ is 0, this simplifies to $a_{ii} = -c_{ii} X_i^*$.

In order to test the potential stability of our observed networks, for analytical purposes, we assumed that the observed systems were at or near a steady state. This meant that we could equate the energy loss rate for each species interaction $f_{ij}$ (see above) to the competition term in the differential equation at equilibrium, so that $f_{ij} \approx -c_{ij} X_i^* X_j^*$. The observed abundances $B_i$, which capture the number of colonies of species $i$ in a given assemblage, were equated with the equilibrium density $X^*$. The elements of the Jacobian matrix could thus be calculated as

$$a_{ij} = -c_{ij} X_i^* = \frac{-c_{ij} X_i^* X_j^*}{X_j^*} = \frac{f_{ij}}{B_j} \qquad (2)$$

**Constructing asymmetric and symmetric theoretical matrices.** Following ref. [23], we constructed competitive community matrices with random topology and interaction strengths, varying the number of species $S$ and connectance $C$. Diagonal values were set to $-1$. Interaction strengths $a_{ij}$ were drawn from a normal distribution centred on 0 and a standard deviation of $\sigma = -0.8$. To ensure that all interactions represented interference competition, the negative of the absolute value of each random number was used.

We manipulated the degree of asymmetry of pairwise interactions in the random networks by sorting the randomly drawn interaction strengths into pairs. For the random distribution, the value of each link was drawn independently. For asymmetric pairs, we first determined the number of links $n$ and then drew a list of $2n$ random interaction strengths. After ordering the list from largest to weakest interaction strength, we sorted them into the matrix by pairing the strongest link to the weakest, the second strongest to the second weakest, etc. In symmetric matrices, we did the opposite and assigned the values as they were ordered in the list. Thus, the strongest link was paired with the second strongest, the third with the fourth, etc.

**Calculating stability.** The stability of a system is the ability to return to its original state after a disturbance. For a system represented by Jacobian matrix $A$, it is calculated by determining the eigenvalues $\lambda$ of the matrix A[23]. The real parts of the eigenvalues, $Re(\lambda)$, describe the rate of exponential growth or decay with which a small perturbation would increase (if the rate is positive) or decrease (if it is negative). An equilibrium is only considered stable if all eigenvalues have negative real parts, meaning that all perturbations decay over time. The so-called dominant eigenvalue $\lambda_d$ (that is, the eigenvalue with the largest real part) is thus an indicator of system stability: If the real part of $\lambda_d$ is negative, the system is stable since the real parts of all other eigenvalues must be negative as well.

$Re(\lambda_d)$ of the Jacobian has the dimension per time. This means that it is not suitable to compare the stability properties of different systems encompassing different time scales. To be able to make this comparison and in order to assess how far the systems are from stability, we use the dimensionless metric $s^*$, the critical level of self-regulation, following previous work on food webs[26]. $s^*$ represents the factor by which observed intraspecific interaction strength has to be

multiplied to obtain stability. The metric $s^*$ thus describes how far the system is from stability, as a multiplier of the actual amount of self-regulation, in the case of an unstable system ($s^* > 1$). In the case of a stable system ($s^* < 1$), it indicates how much 'buffering capacity' a system has, giving the fraction of observed self-regulation that is enough to provide stability. Determining $s^*$ can be done numerically by multiplying the diagonal of the Jacobian matrix with a control parameter $s$ and adjusting $s$ until the matrix is right at the threshold between stability and instability.

Our empirical data did not always contain estimates of the strength of self-regulation for all species. This was because intraspecific competition is often rare in species with low abundances, and therefore no contests could be observed during data collection. This does not mean that those populations are not self-regulated at all, just that the area of collection was not big enough to observe it. For our relative stability analysis, diagonal elements that could not be estimated from observational data were set to a value that was proportional to the mean of all Jacobian elements in the matrix. Replacing the zero self-regulation terms in the original Jacobian matrices with small non-zero negative values did not change the result that all systems were unstable. The exact proportionality factor affected the absolute values of both $s^*$ and $w^{(2)}_{max}$, but it did not affect the resulting stability pattern (Supplementary Figs. 6 and 7). The results in the main analysis were carried out using a proportionality factor of 0.1.

**Normalisation procedure.** For our manipulation experiments with the observed systems, we normalised the Jacobian matrices, following ref. [25]. The normalisation procedure translated the diagonal structure of matrix A onto the off-diagonal structure. This meant that manipulations of the off-diagonal elements could be carried out without affecting the diagonal structure, making the feedback- and stability analysis more transparent.

The Jacobian matrices were normalised by dividing each row in the matrix by the absolute value of its corresponding diagonal term. The resulting matrix $\bar{A}$, which we call "community matrix", has the dimensionless elements $\bar{a}_{ij} = \frac{a_{ij}}{|a_{ii}|}$ and a uniform diagonal of $-1$. We could then calculate $s^*$ of this matrix $\bar{A}$ directly by setting the diagonals of $\bar{A}$ at 0, obtaining $\bar{A}_0$, and calculating the maximum real part of the eigenvalues of the matrix $\bar{A}_0$. Under certain conditions, this maximum real part, the critical level of self-regulation $s^*$ of a normalised matrix, equals the $s^*$ of the original Jacobian[30]. We found that for all our observed systems, the numerically determined $s^*$ of the original Jacobian matrix A was the same as or was approximated very closely by the real part of the dominant eigenvalue of the normalised community matrix with diagonals set at zero (Supplementary Table 2). This meant that normalising our observed systems maintained the feedback structure and essential stability properties.

**Calculating loop weight.** Feedback loops are closed chains of interactions. They are quantified as the product of all link strengths within them. Thus, in competition networks, where all $a_{ij} < 0$, all loops with an even number of links are positive, while those with an odd number of links are negative. Isolated positive feedback loops amplify perturbations and are thus generally seen as destabilising, while isolated negative loops are seen as stabilising as they dampen perturbations[21]. Together, the positive and negative feedback loops regulate a whole system.

Whether a complex system is stable is determined by the multitude of strengths of negative and positive feedback loops, and the relation between individual feedback loops and system stability is not straightforward[21]. Systems of $n$ components comprise feedback loops of various lengths: 2-link, 3-link, 4-link and so on, up to length $n$. The total feedback at any given level $k$ is a summation of the strengths of all the feedback loops of length $k$ and that of all the combinations of disjunct (non-overlapping) feedback loops of shorter length containing $k$ elements. A necessary (but not sufficient) condition for stability states that in a system of $n$ variables, the total feedback $F_k$ for each level $k$ in the system must be negative[21]. We used the quantity loop weight $w^{(k)}$, the geometric mean of all links in a feedback loop of length $k$[24], scaled to their respective self-regulation terms:

$$w^{(k)} = \left| \frac{a_{i_1 i_2} a_{i_2 i_3} \cdots a_{i_k i_1}}{a_{i_1 i_1} a_{i_2 i_2} \cdots a_{i_k i_k}} \right|^{1/k},$$

to be able to compare feedback strength with $s^*$ and compare loops of different lengths.

**Calculation of transitivity indices and asymmetry measures.** We calculated two known indices that are applicable to our dataset:

- Gallien's intransitivity index[28] allows the inclusion of reciprocal links, but its calculation is not straightforward for larger networks, as the index values of all intransitive structures have to be summed up. The index is designed for competition coefficients. As we do not have those, we calculated the index using the elements of the Jacobian matrix.
- Tanaka and Nandakumar's transitivity index[27] can be directly calculated from the species-contact matrices. However, this index only measures the asymmetry (or polarisation) of pairwise interactions.

To quantify the patterns in interaction strengths that emerge from the hierarchical structure, we used two new indices to characterise the structure of competition networks. They can be applied to any matrix and were used to determine asymmetries in energy loss rates, interaction strengths as well as normalised interaction strengths

1. We quantified *pairwise asymmetry* as the ratio of the stronger and the weaker value for each pair of interaction strengths $a_{ij}a_{ji}$. These ratios were then averaged over all pairs of interaction strengths to get the pairwise asymmetry of the whole network:

$$\text{pairwise asymmetry} = \frac{1}{n}\sum_{i,j=1}^{n}\frac{a_{ij}}{a_{ji}} \qquad (3)$$

   with $n$ = number of pairwise interactions and $a_{ij} > a_{ji}$

2. The hierarchical ranking of the species causes what we call *community asymmetry*—strong differences between strong "top-down" effects from higher-ranked species to lower-ranked species and weak "bottom-up" effects from lower-ranked species to higher-ranked ones. If the species in the community matrix $A$ are ordered according to their hierarchical ranking, all strong values are located below the diagonal, in the lower triangle, while all weaker values are located in the upper triangle. We defined community asymmetry of a matrix as the highest ratio, given any ordering of species, of the mean of all non-zero interaction strengths located below the diagonal and the mean of all non-zero interaction strengths below the diagonal

$$\text{community asymmetry} = \frac{mean(a_{ij})}{mean(a_{ji})} \qquad (4)$$

   with $i > j$. To determine community asymmetries of our systems, we simply determined this ratio for all possible orders of species and then chose the one that maximised the ratio.

**Manipulating empirical network structure.** We manipulated the internal organisation of the raw and normalised empirical systems in order to destroy or add specific structures of interest. During each manipulation, the non-zero off-diagonal elements were reshuffled within the matrix. The manipulations meant that both the complexity and the qualitative structure of the network remained intact, and all interaction strength values (hence also mean interaction strength) were preserved. For the normalised matrices, the relation between diagonal and off-diagonal values was not affected by the manipulations since all diagonal values of the normalised matrices had the value $-1$.

We used three types of randomisation procedures to destroy the observed structure:

- full randomisation: all non-zero interspecific interaction strengths were randomly reshuffled within the network. This procedure destroyed the loop-weight structure of all feedback loops of length >1.
- weak randomisation: non-zero pairs of interaction strengths were reshuffled within the network. Thus, the 2-link loops were kept intact but their location within the network was randomised. This disrupted patterns associated with feedback loops of length >2.
- minimal randomisation: identical to weak randomisation but with preserved above/below diagonal orientation of the pairs of matrix elements.

Additionally, we used two types of manipulations on the fully randomised matrices to artificially restore the observed structure:

- Adding pairwise asymmetry: the pairwise asymmetry of interactions was maximised by pairing the strongest element of the matrix with the weakest, the second strongest with the second weakest (following the same procedure that was used for the theoretical matrices). The location of the pairs was randomised in the system so that strong links could appear on either side of the diagonal.
- Adding community asymmetry on top of pairwise asymmetry: with a pairwise asymmetry back in place, the pairs were placed so that a stronger value of each interaction was put below the matrix diagonal. This maximised the community asymmetry for the given set of interaction strengths.

**Reporting summary**. Further information on research design is available in the Nature Portfolio Reporting Summary linked to this article.

## Data availability
All empirical data used in this study have been deposited at the public repository Zenodo (https://doi.org/10.5281/zenodo.8010451)[50] and can be accessed under the following URL: https://zenodo.org/record/8010451.

## Code availability
The analysis was performed in R (version 4.3.0) and Python 3. All script files needed to reproduce the analysis, including all figures and tables, are available in the public

repository Zenodo (https://doi.org/10.5281/zenodo.8010451)[50] and can be accessed under the following URL: https://zenodo.org/record/8010451.

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

## Acknowledgements
We thank Jamie Oliver for assistance with graphics; Lloyd Peck and Peter de Ruiter for comments on the manuscript; Jeff Huisman, Richard Law, Nicolas Loeuille, Elisa Thébault, Thilo Gross, Barbara Drossel, Merav Seifan and Jon Pitchford for discussions and all in the Rothera Research Station diving team over the years. This project was funded by the Deutsche Forschungsgemeinschaft (DFG, German Research Foundation) under project number 451967415 (AL 2563/1-1) (F.K., K.T.A.).

## Author contributions
Conceptualisation: A.M.N., K.T.A. Methodology: D.K.A.B., A.M.N., K.T.A., F.K. Software: F.K. Formal analysis: A.M.N., K.T.A., F.K. Resources: D.K.A.B., K.T. Investigation: D.K.A.B., A.M.N., K.T.A., F.K. Funding acquisition: K.T.A., A.M.N. Supervision: K.T.A. Writing—original draft: A.M.N., K.T.A., F.K. Writing—review & editing: A.M.N., K.T.A., F.K., D.K.A.B., C.Z., K.T.

## Funding

## Competing interests
The authors declare no competing interests.
