## [Peer Review File · Communications Biology]

This manuscript has been previously reviewed at another Nature Portfolio journal. This document only contains reviewer comments and rebuttal letters for versions considered at Communications Biology.

REVIEWERS' COMMENTS:

Reviewer #1 (Remarks to the Author):

This revised version has tested model stability of 30 multispecies bryozoan assemblages, finding that competitive hierarchies mitigate network instability. I think this finding is novel. When I read the authors' responses to my previous comments, I think they have addressed all of them sufficiently. As such, I would like to recommend it for publication in Communications Biology.

Reviewer #2 (Remarks to the Author):

I found that the paper presents robust evidence to support the hypothesis that competitive hierarchy does play an important role in the stability of coexistence benthic marine communities.

My main concern is related to the organization of the text. There are several sentences and, in some cases, full paragraphs that are misplaced. I found the Introduction section confusing in the sense that there are details regarding the methodology used in between some paragraphs. For instance, L 41-44, 51-53, 62-69, 70-76. The Introduction should serve as the entrance to the topic you are willing to make a step forward to in the state of knowledge. At the end of this section, the authors should clearly mention the objectives of the research.

This misplacement also happens in Results, e.g. L 99-108, 169-176, 183-189, L205-206 correspond to Methods. L 113-128 correspond to Introduction.

The concluding paragraph seems to go way beyond the analyses performed and, mostly, the scope and objective of the research. I suggest to keep the conclusions within the framework of Ecology.

Minor comments:

L 45. The correct term is 'iceberg', not 'ice-berg'.

L 80-81. '... where connectance is the number of ...'. This corresponds to Methods.

L 113-128. Please provide references to support this.

L 137. Please be more specific on the sensitivity analysis.

L 142-149. This is redundant with L 350-361 of Methods and should be deleted from Results.

L 150-156. This is redundant with Methods subsection 'Calculating loop weight'.

L 162. I suggest to replace 'interestingly' by 'it's noteworthy that'. The term interestingly is subjective and cannot be justified in Results. It might be used in Discussion because there you can justify why it is interesting.

L 166. Correct 'selfregulation'.

L 267-269. Please add latitude and longitude coordinates and time period for the study areas considered.

L 327-328. It is not clear what it is meant by '... with a specified number of species S and connectance C.' If this refers to conserving the same topology as the empirical networks in order to be able to compare results, please clarify it.

L 342-349. Please add references supporting the use of the Jacobian matrix as a way of estimating system's stability.

L 364. Replace 'confrontations' by 'contests'.

L 394. I suggest to replace '2,4,6...' and '3,5,7...' by 'even' and 'odd', respectively.

L 395-398. Please provide references to support this.

L 399. I suggest: 'Whether a system is stable, meaning the ability to return to its original...'

L 400. Please provide a reference supporting this.

Responses to comments by reviewer 2

I found that the paper presents robust evidence to support the hypothesis that competitive hierarchy does play an important role in the stability of coexistence benthic marine communities.

We thank the reviewer for this positive feedback.

My main concern is related to the organization of the text. There are several sentences and, in some cases, full paragraphs that are misplaced. I found the Introduction section confusing in the sense that there are details regarding the methodology used in between some paragraphs. For instance, L 41-44, 51-53, 62-69, 70-76. The Introduction should serve as the entrance to the topic you are willing to make a step forward to in the state of knowledge. At the end of this section, the authors should clearly mention the objectives of the research.

The reviewer is correct in pointing out that the previous version of the introduction contained too many technical details that would be better placed in the methods sections and we therefore modified our text accordingly. In particular, we shortened lines 41/44 (now in lines 66/67), 53/54 (now in lines 75/76) and removed lines 63-69. We also removed methodological details from the last paragraph and state our research objectives more clearly (now in lines 86-92, this includes what was previously lines 70-76).

This misplacement also happens in Results, e.g. L 99-108, 169-176, 183-189, L205-206 correspond to Methods.

We removed several methodological explanations but kept a few sentences that we believe are important for understanding the theoretical rationale. For lines 99-108, 169-176 and 183-198 we shortened the text (now in lines 113-117; 156-160; 169-174). We kept the sentence in lines 205-206 because it was essential for making the argument clear.

L 113-128 correspond to Introduction.

We moved the explanation of the competitive 2-species system from the results section into the introduction (now in lines 32-39 and 54-63).

The concluding paragraph seems to go way beyond the analyses performed and, mostly, the scope and objective of the research. I suggest to keep the conclusions within the framework of Ecology.

We agree with the referee's comment and therefore rephrased the final paragraph (previously lines 261-264, now in lines 244-252).

Minor comments:

L 45. The correct term is 'iceberg', not 'ice-berg'.

We changed the term 'ice-berg scour' to 'ice scour' (now in line 68).

L 80-81. '... where connectance is the number of ...'. This corresponds to Methods.

We removed this part of the sentence (now in line 96).

L 113-128. Please provide references to support this.

We added a reference to Levins (1974) in what is now line 60.

L 137. Please be more specific on the sensitivity analysis.

We added details on the sensitivity analysis, now in lines 130-135.

L 142-149. This is redundant with L 350-361 of Methods and should be deleted from Results.

We believe that the definition of the critical level of self-regulation is important for the reader to follow the text here. However, we shortened the paragraph and removed what was previously lines 146-149.

L 150-156. This is redundant with Methods subsection 'Calculating loop weight'.

We removed this paragraph.

L 162. I suggest to replace 'interestingly' by 'it's noteworthy that'. The term interestingly is subjective and cannot be justified in Results. It might be used in Discussion because there you can justify why it is interesting.

Done.

L 166. Correct 'selfregulation'.

We corrected selfregulation to self-regulation (now in line 153).

L 267-269. Please add latitude and longitude coordinates and time period for the study areas considered.

We added coordinates for each collection site and year of collection (now in lines 257-262).

L 327-328. It is not clear what it is meant by '... with a specified number of species S and connectance C .' If this refers to conserving the same topology as the empirical networks in order to be able to compare results, please clarify it.

We reformulated this sentence into: "we constructed competitive community matrices with random topology and interaction strengths, varying the number of species S and connectance C ." (now in lines 321-322).

L 342-349. Please add references supporting the use of the Jacobian matrix as a way of estimating system's stability.

We added a reference to May 1972 (now in line 337).

L 364. Replace 'confrontations' by 'contests'.

Done.

L 394. I suggest to replace '2,4,6...' and '3,5,7...' by 'even' and 'odd', respectively.

We changed this to: “all loops with an even number of links are positive, while those with an odd number of links are negative.” (now in lines 387-388).

L 395-398. Please provide references to support this.

We added a reference to Levins 1974 (now in line 391).

L 399. I suggest: ‘Whether a system is stable, meaning the ability to return to its original...’
We edited this sentence as suggested and moved it forward to the section “Calculating stability” (now in lines 335).

L 400. Please provide a reference supporting this.

We added a reference to Levins 1974 (now in line 394).